# OBLIQUE DECISION TREES FROM DERIVATIVES OF ReLU NETWORKS

**Guang-He Lee & Tommi S. Jaakkola**
Computer Science and Artificial Intelligence Lab
MIT
{guanghe,tommi}@csail.mit.edu

## ABSTRACT

We show how neural models can be used to realize piece-wise constant functions such as decision trees. The proposed architecture, which we call locally constant networks, builds on ReLU networks that are piece-wise linear and hence their associated gradients with respect to the inputs are locally constant. We formally establish the equivalence between the classes of locally constant networks and decision trees. Moreover, we highlight several advantageous properties of locally constant networks, including how they realize decision trees with parameter sharing across branching / leaves. Indeed, only $M$ neurons suffice to implicitly model an oblique decision tree with $2^M$ leaf nodes. The neural representation also enables us to adopt many tools developed for deep networks (e.g., DropConnect (Wan et al., 2013)) while implicitly training decision trees. We demonstrate that our method outperforms alternative techniques for training oblique decision trees in the context of molecular property classification and regression tasks.[1]

## 1 INTRODUCTION

Decision trees (Breiman et al., 1984) employ a series of simple decision nodes, arranged in a tree, to transparently capture how the predicted outcome is reached. Functionally, such tree-based models, including random forest (Breiman, 2001), realize piece-wise constant functions. Beyond their status as de facto interpretable models, they have also persisted as the state of the art models in some tabular (Sandulescu & Chiru, 2016) and chemical datasets (Wu et al., 2018). Deep neural models, in contrast, are highly flexible and continuous, demonstrably effective in practice, though lack transparency. We merge these two contrasting views by introducing a new family of neural models that implicitly learn and represent oblique decision trees.

Prior work has attempted to generalize classic decision trees by extending coordinate-wise cuts to be weighted, linear classifications. The resulting family of models is known as oblique decision trees (Murthy et al., 1993). However, the generalization accompanies a challenging combinatorial, non-differentiable optimization problem over the linear parameters at each decision point. Simple sorting procedures used for successively finding branch-wise optimal coordinate cuts are no longer available, making these models considerably harder to train. While finding the optimal oblique decision tree can be cast as a mixed integer linear program (Bertsimas & Dunn, 2017), scaling remains a challenge.

In this work, we provide an effective, implicit representation of piece-wise constant mappings, termed *locally constant networks*. Our approach exploits piece-wise linear models such as ReLU networks as basic building blocks. Linearity of the mapping in each region in such models means that the gradient with respect to the input coordinates is locally constant. We therefore implicitly represent piece-wise constant networks through gradients evaluated from ReLU networks. We prove the equivalence between the class of oblique decision trees and these proposed locally constant neural models. However, the sizes required for equivalent representations can be substantially different. For example, a locally constant network with $M$ neurons can implicitly realize an oblique decision tree whose explicit form requires $2^M - 1$ oblique decision nodes. The exponential complexity reduc-

---

[1]Our implementation and data are available at `https://github.com/guanghelee/iclr20-lcn`.

tion in the corresponding neural representation illustrates the degree to which parameters are shared across the locally constant regions.

Our locally constant networks can be learned via gradient descent, and they can be explicitly converted back to oblique decision trees for interpretability. For learning via gradient descent, however, it is necessary to employ some smooth annealing of piece-wise linear activation functions so as to keep the gradients themselves continuous. Moreover, we need to evaluate the gradients of all the neurons with respect to the inputs. To address this bottleneck, we devise a dynamic programming algorithm which computes all the necessary gradient information in a single forward pass. A number of extensions are possible. For instance, we can construct *approximately* locally constant networks by switching activation functions, or apply helpful techniques used with normal deep learning models (e.g., DropConnect (Wan et al., 2013)) while implicitly training tree models.

We empirically test our model in the context of molecular property classification and regression tasks (Wu et al., 2018), where tree-based models remain state-of-the-art. We compare our approach against recent methods for training oblique decision trees and classic ensemble methods such as gradient boosting (Friedman, 2001) and random forest. Empirically, a locally constant network always outperforms alternative methods for training oblique decision trees by a large margin, and the ensemble of locally constant networks is competitive with classic ensemble methods.

## 2 RELATED WORK

Locally constant networks are built on a mixed integer linear representation of piece-wise linear networks, defined as any feed-forward network with a piece-wise linear activation function such as ReLU (Nair & Hinton, 2010). One can specify a set of integers encoding the active linear piece of each neuron, which is called an activation pattern (Raghu et al., 2017). The feasible set of an activation pattern forms a convex polyhedron in the input space (Lee et al., 2019), where the network degenerates to a linear model. The framework motivates us to leverage the locally invariant derivatives of the networks to construct a locally constant network. The activation pattern is also exploited in literature for other purposes such as deriving robustness certificates (Weng et al., 2018). We refer the readers to the recent work (Lee et al., 2019) and the references therein.

Locally constant networks use the gradients of deep networks with respect to inputs as the representations to build discriminative models. Such gradients have been used in literature for different purposes. They have been widely used for local sensitivity analysis of trained networks (Simonyan et al., 2013; Smilkov et al., 2017). When the deep networks model an energy function (LeCun et al., 2006), the gradients can be used to draw samples from the distribution specified by the normalized energy function (Du & Mordatch, 2019; Song & Ermon, 2019). The gradients can also be used to train generative models (Goodfellow et al., 2014) or perform knowledge distillation (Srinivas & Fleuret, 2018).

The class of locally constant networks is equivalent to the class of oblique decision trees. There are some classic methods that also construct neural networks that reproduce decision trees (Sethi, 1990; Brent, 1991; Cios & Liu, 1992), by utilizing step functions and logic gates (e.g., AND/NEGATION) as the activation function. The methods were developed when back-propagation was not yet practically useful, and the motivation is to exploit effective learning procedures of decision trees to train neural networks. Instead, our goal is to leverage the successful deep models to train oblique decision trees. Recently, Yang et al. (2018) proposed a network architecture with $\arg\max$ activations to represent classic decision trees with coordinate cuts, but their parameterization scales exponentially with input dimension. In stark contrast, our parameterization only scales linearly with input dimension (see our complexity analyses in §3.7).

Learning oblique decision trees is challenging, even for a greedy algorithm; for a single oblique split, there can be $\sum_{k=0}^{D}\binom{N}{k}$ different ways to separate $N$ data points in $D$-dimensional space (Vapnik & Chervonenkis, 1971) (cf. $ND$ possibilities for coordinate-cuts). Existing learning algorithms for oblique decision trees include greedy induction, global optimization, and iterative refinements on an initial tree. We review some representative works, and refer the readers to the references therein.

Optimizing each oblique split in greedy induction can be realized by coordinate descent (Murthy et al., 1994) or a coordinate-cut search in some linear projection space (Menze et al., 2011; Wickramarachchi et al., 2016). However, the greedy constructions tend to get stuck in poor local optimum.

There are some works which attempt to find the global optimum given a fixed tree structure by formulating a linear program (Bennett, 1994) or a mixed integer linear program (Bertsimas & Dunn, 2017), but the methods are not scalable to ordinary tree sizes (e.g., depth more than 4). The iterative refinements are more scalable than global optimization, where CART (Breiman et al., 1984) is the typical initialization. Carreira-Perpinán & Tavallali (2018) develop an alternating optimization method via iteratively training a linear classifier on each decision node, which yield the state-of-the-art empirical performance, but the approach is only applicable to classification problems. Norouzi et al. (2015) proposed to do gradient descent on a sub-differentiable upperbound of tree prediction errors, but the gradients with respect to oblique decision nodes are unavailable whenever the upperbound is tight. In contrast, our method conducts gradient descent on a differentiable relaxation, which is gradually annealed to a locally constant network.

## 3 METHODOLOGY

In this section, we introduce the notation and basics in §3.1, construct the locally constant networks in §3.2-3.3, analyze the networks in §3.4-3.5, and develop practical formulations and algorithms in §3.6-3.7. Note that we will propose two (equivalent) architectures of locally constant networks in §3.3 and §3.6, which are useful for theoretical analyses and practical purposes, respectively.

### 3.1 NOTATION AND BASICS

The proposed approach is built on feed-forward networks that yield piece-wise linear mappings. Here we first introduce a canonical example of such networks, and elaborate its piece-wise linearity. We consider the densely connected architecture (Huang et al., 2017), where each hidden layer takes as input all the previous layers; it subsumes other existing feed-forward architectures such as residual networks (He et al., 2016). For such a network $f_\theta : \mathbb{R}^D \to \mathbb{R}^L$ with the set of parameters $\theta$, we denote the number of hidden layers as $M$ and the number of neurons in the $i^{\text{th}}$ layer as $N_i$; we denote the neurons in the $i^{\text{th}}$ layer, before and after activation, as $\boldsymbol{z}^i \in \mathbb{R}^{N_i}$ and $\boldsymbol{a}^i \in \mathbb{R}^{N_i}$, respectively, where we sometimes interchangeably denote the input instance $\boldsymbol{x}$ as $\boldsymbol{a}^0 \in \mathbb{R}^{N_0}$ with $N_0 \triangleq D$. To simplify exposition, we denote the concatenation of $(\boldsymbol{a}^0, \boldsymbol{a}^1, \dots, \boldsymbol{a}^i)$ as $\tilde{\boldsymbol{a}}^i \in \mathbb{R}^{\tilde{N}_i}$ with $\tilde{N}_i \triangleq \sum_{j=0}^{i} N_i, \forall i \in \{0, 1, \dots, M\}$. The neurons are defined via the weight matrix $\boldsymbol{W}^i \in \mathbb{R}^{N_i \times \tilde{N}_{i-1}}$ and the bias vector $\boldsymbol{b}^i \in \mathbb{R}^{N_i}$ in each layer $i \in [M] \triangleq \{1, 2, \dots, M\}$. Concretely,

$$\boldsymbol{a}^0 \triangleq \boldsymbol{x}, \quad \boldsymbol{z}^i \triangleq \boldsymbol{W}^i \tilde{\boldsymbol{a}}^{i-1} + \boldsymbol{b}^i, \quad \boldsymbol{a}^i \triangleq \sigma(\boldsymbol{z}^i), \forall i \in [M], \tag{1}$$

where $\sigma(\cdot)$ is a point-wise activation function. Note that both $\boldsymbol{a}$ and $\boldsymbol{z}$ are functions of the specific instance denoted by $\boldsymbol{x}$, where we drop the functional dependency to simplify notation. We use the set $\mathcal{I}$ to denote the set of all the neuron indices in this network $\{(i, j) | j \in [N_i], i \in [M]\}$. In this work, we will use ReLU (Nair & Hinton, 2010) as a canonical example for the activation function

$$\boldsymbol{a}_j^i = \sigma(\boldsymbol{z}^i)_j \triangleq \max(0, \boldsymbol{z}_j^i), \forall (i, j) \in \mathcal{I}, \tag{2}$$

but the results naturally generalize to other piece-wise linear activation functions such as leaky ReLU (Maas et al., 2013). The output of the entire network $f_\theta(\boldsymbol{x})$ is the affine transformation from all the hidden layers $\tilde{\boldsymbol{a}}^M$ with the weight matrix $\boldsymbol{W}^{M+1} \in \mathbb{R}^{L \times \tilde{N}_M}$ and bias vector $\boldsymbol{b}^{M+1} \in \mathbb{R}^L$.

### 3.2 LOCAL LINEARITY

It is widely known that the class of networks $f_\theta(\cdot)$ yields a piece-wise linear function. The results are typically proved via associating the end-to-end behavior of the network with its activation pattern – which linear piece in each neuron is activated; once an activation pattern is fixed across the entire network, the network degenerates to a linear model and the feasible set with respect to an activation pattern is a natural characterization of a locally linear region of the network.

Formally, we define the activation pattern as the collection of activation indicator functions for each neuron $\boldsymbol{o}_j^i : \mathbb{R}^D \to \{0, 1\}, \forall (i, j) \in \mathcal{I}$ (or, equivalently, the derivatives of ReLU units; see below)[2]:

$$\boldsymbol{o}_j^i = \frac{\partial \boldsymbol{a}_j^i}{\partial \boldsymbol{z}_j^i} \triangleq \mathbb{I}[\boldsymbol{z}_j^i \geq 0], \forall (i, j) \in \mathcal{I}, \tag{3}$$

---

[2]Note that each $\boldsymbol{o}_j^i$ is again a function of $\boldsymbol{x}$, where we omit the dependency for brevity.

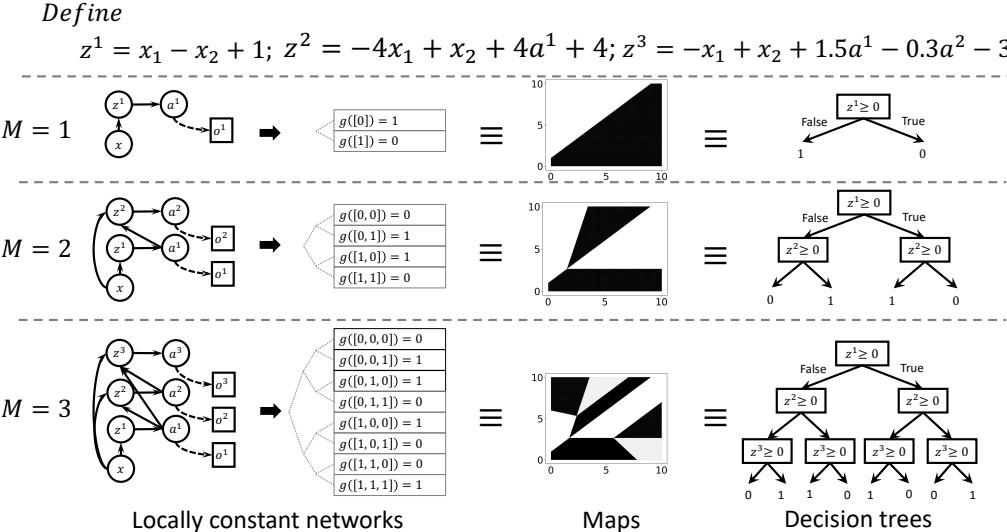

Figure 1: Toy examples for the equivalent representations of the same mappings for different $M$. Here the locally constant networks have 1 neuron per layer. We show the locally constant networks on the LHS, the raw mappings in the middle, and the equivalent oblique decision trees on the RHS.

where $\mathbb{I}[\cdot]$ is the indicator function. Note that, for mathematical correctness, we *define* $\partial \boldsymbol{a}_j^i / \partial \boldsymbol{z}_j^i = 1$ at $\boldsymbol{z}_j^i = 0$; this choice is arbitrary, and one can change it to $\partial \boldsymbol{a}_j^i / \partial \boldsymbol{z}_j^i = 0$ at $\boldsymbol{z}_j^i = 0$ without affecting most of the derivations. Given a *fixed* activation pattern $\bar{\boldsymbol{o}}_j^i \in \{0, 1\}, \forall (i, j)$, we can specify a feasible set in $\mathbb{R}^D$ that corresponds to this activation pattern $\{\boldsymbol{x} \in \mathbb{R}^D | \boldsymbol{o}_j^i = \bar{\boldsymbol{o}}_j^i, \forall (i, j) \in \mathcal{I}\}$ (note that each $\boldsymbol{o}_j^i$ is a function of $\boldsymbol{x}$). Due to the fixed activation pattern, the non-linear ReLU can be re-written as a *linear* function for all the inputs *in the feasible set*. For example, for an $\bar{\boldsymbol{o}}_j^i = 0$, we can re-write $\boldsymbol{a}_j^i = 0 \times \boldsymbol{z}_j^i$. As a result, the network has a consistent end-to-end linear behavior across the entire feasible set. One can prove that all the feasible sets partition the space $\mathbb{R}^D$ into disjoint convex polyhedra[3], which realize a natural representation of the locally linear regions. Since we will only use the result to motivate the construction of locally constant networks, we refer the readers to Lee et al. (2019) for a detailed justification of the piece-wise linearity of such networks.

### 3.3 CANONICAL LOCALLY CONSTANT NETWORKS

Since the ReLU network $f_\theta(\boldsymbol{x})$ is piece-wise linear, it immediately implies that its derivatives with respect to the input $\boldsymbol{x}$ is a piece-wise constant function. Here we use $J_{\boldsymbol{x}} f_\theta(\boldsymbol{x}) \in \mathbb{R}^{L \times D}$ to denote the Jacobian matrix (i.e., $[J_{\boldsymbol{x}} f_\theta(\boldsymbol{x})]_{i,j} = \partial f_\theta(\boldsymbol{x})_i / \partial \boldsymbol{x}_j$), and we assume the Jacobian is consistent with Eq. (3) at the boundary of the locally linear regions. Since any function taking the piece-wise constant Jacobian as input will remain itself piece-wise constant, we can construct a variety of locally constant networks by composition.

However, in order to simplify the derivation, we first make a trivial observation that the activation pattern in each locally linear region is also locally invariant. More broadly, any invariant quantity in each locally linear region can be utilized so as to build locally constant networks. We thus define the locally constant networks as any composite function that leverage the local invariance of piece-wise linear networks. For the theoretical analyses, we consider the below architecture.

**Canonical architecture.** We denote $\tilde{\boldsymbol{o}}^M \in \{0, 1\}^{\tilde{N}_M}$ as the concatenation of $(\boldsymbol{o}^1, \dots, \boldsymbol{o}^M)$. We will use the composite function $g(\tilde{\boldsymbol{o}}^M)$ as the canonical architecture of locally constant networks for theoretical analyses, where $g : \{0, 1\}^{\tilde{N}_M} \to \mathbb{R}^L$ is simply a table.

Before elucidating on the representational equivalence to oblique decision trees, we first show some toy examples of the canonical locally constant networks and their equivalent mappings in Fig. 1,

---

[3]The boundary of the polyhedron depends on the specific definition of the activation pattern, so, under some definition in literature, the resulting convex polyhedra may not be disjoint in the boundary.

which illustrates their constructions when there is only $1$ neuron per layer (i.e., $\boldsymbol{z}^i = \boldsymbol{z}^i_1$, and similarly for $\boldsymbol{o}^i$ and $\boldsymbol{a}^i$). When $M = 1$, $\boldsymbol{o}^1 = 1 \Leftrightarrow \boldsymbol{x}_1 - \boldsymbol{x}_2 + 1 \geq 0$, thus the locally constant network is equivalent to a linear model shown in the middle, which can also be represented as an oblique decision tree with depth $= 1$. When $M > 1$, the activations in the previous layers control different linear behaviors of a neuron with respect to the input, thus realizing a hierarchical structure as an oblique decision tree. For example, for $M = 2$, $\boldsymbol{o}^1 = 0 \Leftrightarrow \boldsymbol{z}^1 < 0 \Rightarrow \boldsymbol{z}^2 = -4\boldsymbol{x}_1 + \boldsymbol{x}_2 + 4$ and $\boldsymbol{o}^1 = 1 \Leftrightarrow \boldsymbol{z}^1 \geq 0 \Rightarrow \boldsymbol{z}^2 = -3\boldsymbol{x}_2 + 8$; hence, it can also be interpreted as the decision tree on the RHS, where the *concrete realization* of $\boldsymbol{z}^2$ depends on the previous decision variable $\boldsymbol{z}^1 \geq 0$. Afterwards, we can map either the activation patterns on the LHS or the decision patterns on the RHS to an output value, which leads to the mapping in the middle.

## 3.4 REPRESENTATIONAL EQUIVALENCE

In this section, we prove the equivalence between the class of oblique decision trees and the class of locally constant networks. We first make an observation that any unbalanced oblique decision tree can be re-written to be balanced by adding dummy decision nodes $\boldsymbol{0}^\top \boldsymbol{x} \geq -1$. Hence, we can define the *class* of oblique decision trees with the balance constraint:

**Definition 1.** *The class of oblique decision trees contains any functions that can be procedurally defined (with some depth $T \in \mathbb{Z}_{>0}$) for $\boldsymbol{x} \in \mathbb{R}^D$:*

1. $\boldsymbol{r}_1 \triangleq \mathbb{I}[\boldsymbol{\omega}_\varnothing^\top \boldsymbol{x} + \beta_\varnothing \geq 0]$, *where* $\boldsymbol{\omega}_\varnothing \in \mathbb{R}^D$ *and* $\beta_\varnothing \in \mathbb{R}$ *denote the weight and bias of the root decision node.*

2. *For* $i \in (2, 3, \ldots, T)$, $\boldsymbol{r}_i \triangleq \mathbb{I}[\boldsymbol{\omega}_{\boldsymbol{r}_{1:i-1}}^\top \boldsymbol{x} + \beta_{\boldsymbol{r}_{1:i-1}} \geq 0]$, *where* $\boldsymbol{\omega}_{\boldsymbol{r}_{1:i-1}} \in \mathbb{R}^D$ *and* $\beta_{\boldsymbol{r}_{1:i-1}} \in \mathbb{R}$ *denote the weight and bias for the decision node after the decision pattern* $\boldsymbol{r}_{1:i-1}$.

3. $v : \{0, 1\}^T \rightarrow \mathbb{R}^L$ *outputs the leaf value* $v(\boldsymbol{r}_{1:T})$ *associated with the decision pattern* $\boldsymbol{r}_{1:T}$.

The class of locally constant networks is defined by the *canonical architecture* with finite $M$ and $N_i, \forall i \in [M]$. We first prove that we can represent any oblique decision tree as a locally constant network. Since a typical oblique decision tree can produce an arbitrary weight in each decision node (cf. the structurally dependent weights in the oblique decision trees in Fig. 1), the idea is to utilize a network with only $1$ hidden layer such that the neurons do not constrain one another. Concretely,

**Theorem 2.** *The class of locally constant networks $\supseteq$ the class of oblique decision trees.*

*Proof.* For any oblique decision tree with depth $T$, it contains $2^T - 1$ weights and biases. We thus construct a locally constant network with $M = 1$ and $N_1 = 2^T - 1$ such that each pair of $(\boldsymbol{\omega}, \beta)$ in the oblique decision tree is equal to some $\boldsymbol{W}^1_{k,:}$ and $\boldsymbol{b}^1_k$ in the constructed locally constant network.

For each leaf node in the decision tree, it is associated with an output value $\boldsymbol{y} \in \mathbb{R}^L$ and $T$ decisions; the decisions can be written as $\boldsymbol{W}^1_{\text{idx}[j],:}\boldsymbol{x} + \boldsymbol{b}^1_{\text{idx}[j]} \geq 0$ for $j \in \{1, 2, \ldots, T'\}$ and $\boldsymbol{W}^1_{\text{idx}[j],:}\boldsymbol{x} + \boldsymbol{b}^1_{\text{idx}[j]} < 0$ for $j \in \{T' + 1, T' + 2, \ldots, T\}$ for some index function $\text{idx} : [T] \rightarrow [2^T - 1]$ and some $T' \in \{0, 1, \ldots, T\}$. We can set the table $g(\cdot)$ of the locally constant network as

$$\boldsymbol{y}, \text{ if } \begin{cases} \boldsymbol{o}^1_{\text{idx}[j]} = 1(\Leftrightarrow \boldsymbol{W}^1_{\text{idx}[j],:}\boldsymbol{x} + \boldsymbol{b}^1_{\text{idx}[j]} \geq 0), \text{for } j \in \{1, 2, \ldots, T'\}, \text{ and} \\ \boldsymbol{o}^1_{\text{idx}[j]} = 0(\Leftrightarrow \boldsymbol{W}^1_{\text{idx}[j],:}\boldsymbol{x} + \boldsymbol{b}^1_{\text{idx}[j]} < 0), \text{for } j \in \{T' + 1, T' + 2, \ldots, T\}. \end{cases}$$

As a result, the constructed locally constant network yields the same output as the given oblique decision tree for all the inputs that are routed to each leaf node, which concludes the proof. $\square$

Then we prove that the class of locally constant networks is a subset of the class of oblique decision trees, which simply follows the construction of the toy examples in Fig. 1.

**Theorem 3.** *The class of locally constant networks $\subseteq$ the class of oblique decision trees.*

*Proof.* For any locally constant network, it can be re-written to have $1$ neuron per layer, by expanding any layer with $N_i > 1$ neurons to be $N_i$ different layers such that they do not have effective intra-connections. Below the notation refers to the converted locally constant network with $1$ neuron per layer. We define the following oblique decision tree with $T = M$ for $\boldsymbol{x} \in \mathbb{R}^D$:

1. $\boldsymbol{r}_1 \triangleq \boldsymbol{o}^1_1 = \mathbb{I}[\boldsymbol{\omega}_\varnothing^\top \boldsymbol{x} + \beta_\varnothing \geq 0]$ with $\boldsymbol{\omega}_\varnothing = \boldsymbol{W}^1_{1,:}$ and $\beta_\varnothing = \boldsymbol{b}^1_1$.

2. For $i \in (2, 3, \ldots, M)$, $\boldsymbol{r}_i \triangleq \mathbb{I}[\boldsymbol{\omega}_{\boldsymbol{r}_{1:i-1}}^\top \boldsymbol{x} + \beta_{\boldsymbol{r}_{1:i-1}} \geq 0]$, where $\boldsymbol{\omega}_{\boldsymbol{r}_{1:i-1}} = \nabla_{\boldsymbol{x}} z_1^i$ and $\beta_{\boldsymbol{r}_{1:i-1}} = z_1^i - (\nabla_{\boldsymbol{x}} z_1^i)^\top \boldsymbol{x}$. Note that $\boldsymbol{r}_i = \mathbb{I}[z_1^i \geq 0] = \boldsymbol{o}_1^i$.

3. $v = g$.

Note that, in order to be a valid decision tree, $\boldsymbol{\omega}_{1:\boldsymbol{r}_{i-1}}$ and $\beta_{1:\boldsymbol{r}_{i-1}}$ have to be unique for all $\boldsymbol{x}$ that yield the same decision pattern $\boldsymbol{r}_{1:i-1}$. To see this, for $i \in (2, 3, \ldots, M)$, as $\boldsymbol{r}_{1:i-1} = (\boldsymbol{o}_1^1, \ldots, \boldsymbol{o}_1^{i-1})$, we know each $z_1^i$ is a fixed affine function given an activation pattern for the preceding neurons, so $\nabla_{\boldsymbol{x}} z_1^i$ and $z_1^i - \boldsymbol{x}^\top \nabla_{\boldsymbol{x}} z_1^i$ are fixed quantities given a decision pattern $\boldsymbol{r}_{1:i-1}$.

Since $\boldsymbol{r}_{1:M} = \tilde{\boldsymbol{o}}^M$ and $v = g$, we conclude that they yield the same mapping. □

Despite the simplicity of the proof, it has some practical implications:

**Remark 4.** *The proof of Theorem 3 implies that we can train a locally constant network with $M$ neurons, and convert it to an oblique decision tree with depth $M$ (for interpretability).*

**Remark 5.** *The proof of Theorem 3 establishes that, given a fixed number of neurons, it suffices (representationally) to only consider the locally constant networks with one neuron per layer.*

Remark 5 is important for learning small locally constant networks (which can be converted to shallow decision trees for interpretability), since representation capacity is critical for low capacity models. In the remainder of the paper, we will only consider the setting with $N_i = 1, \forall i \in [M]$.

### 3.5 STRUCTURALLY SHARED PARAMETERIZATION

Although we have established the exact *class-level* equivalence between locally constant networks and oblique decision trees, once we restrict the depth of the locally constant networks $M$, it can no longer re-produce all the decision trees with depth $M$. The result can be intuitively understood by the following reason: we are effectively using $M$ pairs of (weight, bias) in the locally constant network to implicitly realize $2^M - 1$ pairs of (weight, bias) in the corresponding oblique decision tree. Such exponential reduction on the effective parameters in the representation of oblique decision trees yields "dimension reduction" of the model capacity. This section aims to reveal the implied shared parameterization embedded in the oblique decision trees derived from locally constant networks.

In this section, the oblique decision trees and the associated parameters refer to *the decision trees obtained via the proof of Theorem 3*. We start the analysis by a decomposition of $\boldsymbol{\omega}_{\boldsymbol{r}_{1:i}}$ among the preceding weights $\boldsymbol{\omega}_{\varnothing}, \boldsymbol{\omega}_{\boldsymbol{r}_{1:1}}, \ldots, \boldsymbol{\omega}_{\boldsymbol{r}_{1:r-1}}$. To simplify notation, we denote $\boldsymbol{\omega}_{\boldsymbol{r}_{1:0}} \triangleq \boldsymbol{\omega}_{\varnothing}$. Since $\boldsymbol{\omega}_{\boldsymbol{r}_{1:i}} = \nabla_{\boldsymbol{x}} z_1^{i+1}$ and $z_1^{i+1}$ is an affine transformation of the vector $(\boldsymbol{a}_0, \boldsymbol{a}_1^1, \ldots, \boldsymbol{a}_1^i)$,

$$\boldsymbol{\omega}_{\boldsymbol{r}_{1:i}} = \nabla_{\boldsymbol{x}} z_1^{i+1} = \boldsymbol{W}_{1,1:D}^{i+1} + \sum_{k=1}^{i} \boldsymbol{W}_{1,D+k}^{i+1} \times \frac{\partial \boldsymbol{a}_1^k}{\partial \boldsymbol{z}_1^k} \times \nabla_{\boldsymbol{x}} z_1^k = \boldsymbol{W}_{1,1:D}^{i+1} + \sum_{k=1}^{i} \boldsymbol{W}_{1,D+k}^{i+1} \times \boldsymbol{r}_k \times \boldsymbol{\omega}_{\boldsymbol{r}_{1:k-1}},$$

where we simply re-write the derivatives in terms of tree parameters. Since $\boldsymbol{W}_{1,1:D}^{i+1}$ is fixed for all the $\boldsymbol{\omega}_{\boldsymbol{r}_{1:i}}$, the above decomposition implies that, in the induced tree, all the weights $\boldsymbol{\omega}_{\boldsymbol{r}_{1:i}}$ in *the same depth $i$* are restricted to be a linear combination of the fixed basis $\boldsymbol{W}_{1,1:D}^{i+1}$ and the corresponding preceding weights $\boldsymbol{\omega}_{\boldsymbol{r}_{1:0}}, \ldots, \boldsymbol{\omega}_{\boldsymbol{r}_{1:i-1}}$. We can extend this analysis to compare weights in same layer, and we begin the analysis by comparing weights whose $\ell_0$ distance in decision pattern is 1. To help interpret the statement, note that $\boldsymbol{\omega}_{\boldsymbol{r}_{1:j-1}}$ is the weight that leads to the decision $\boldsymbol{r}_j$ (or $\boldsymbol{r}_j'$; see below).

**Lemma 6.** *For an oblique decision tree with depth $T > 1$, $\forall i \in [T-1]$ and any $\boldsymbol{r}_{1:i}, \boldsymbol{r}_{1:i}'$ such that $\boldsymbol{r}_k = \boldsymbol{r}_k'$ for all $k \in [i]$ except that $\boldsymbol{r}_j \neq \boldsymbol{r}_j'$ for some $j \in [i]$, we have*

$$\boldsymbol{\omega}_{\boldsymbol{r}_{1:i}} - \boldsymbol{\omega}_{\boldsymbol{r}_{1:i}'} = \alpha \times \boldsymbol{\omega}_{\boldsymbol{r}_{1:j-1}}, \text{ for some } \alpha \in \mathbb{R}.$$

The proof involves some algebraic manipulation, and is deferred to Appendix A.1. Lemma 6 characterizes an interesting structural constraint embedded in the oblique decision trees realized by locally constant networks, where the structural discrepancy $\boldsymbol{r}_j$ in decision patterns ($\boldsymbol{r}_{1:i}$ versus $\boldsymbol{r}_{1:i}'$) is reflected on the discrepancy of the corresponding weights (up to a scaling factor $\alpha$). The analysis can be generalized for all the weights in the same layer, but the message is similar.

**Proposition 7.** *For the oblique decision tree with depth $T > 1$, $\forall i \in [T-1]$ and any $\boldsymbol{r}_{1:i}, \boldsymbol{r}_{1:i}'$ such that $\boldsymbol{r}_k = \boldsymbol{r}_k'$ for all $k \in [i]$ except for $n \in [i]$ coordinates $j_1, \ldots, j_n \in [i]$, we have*

$$\boldsymbol{\omega}_{\boldsymbol{r}_{1:i}} - \boldsymbol{\omega}_{\boldsymbol{r}_{1:i}'} = \sum_{k=1}^{n} \alpha_k \times \boldsymbol{\omega}_{\boldsymbol{r}_{1:j_k-1}}, \text{ for some } \alpha_k \in \mathbb{R}, \forall k \in [n]. \tag{4}$$

The statement can be proved by applying Lemma 6 multiple times.

**Discussion.** Here we summarize this section and provide some discussion. Locally constant networks implicitly represent oblique decision trees with the same depth and structurally shared parameterization. In the implied oblique decision trees, the weight of each decision node is a linear combination of a shared weight across the whole layer and all the preceding weights. The analysis explains how locally constant networks use only $M$ weights to model a decision tree with $2^M - 1$ decision nodes; it yields a strong regularization effect to avoid overfitting, and helps computation by exponentially reducing the memory consumption on the weights.

### 3.6 STANDARD LOCALLY CONSTANT NETWORKS AND EXTENSIONS

The simple structure of the *canonical* locally constant networks is beneficial for theoretical analysis, but the structure is not practical for learning since the *discrete* activation pattern does not exhibit gradients for learning the networks. Indeed, $\nabla_{\tilde{\boldsymbol{o}}^M} g(\tilde{\boldsymbol{o}}^M)$ is undefined, which implies that $\nabla_{\boldsymbol{W}^i} g(\tilde{\boldsymbol{o}}^M)$ is also undefined. Here we present another architecture that is equivalent to the *canonical* architecture, but exhibits sub-gradients with respect to model parameters and is flexible for model extension.

**Standard architecture.** We assume $N_i = 1, \forall i \in [M]$. We denote the Jacobian of all the neurons after activation $\tilde{\boldsymbol{a}}^M$ as $J_{\boldsymbol{x}} \tilde{\boldsymbol{a}}^M \in \mathbb{R}^{M \times D}$, and denote $\vec{J}_{\boldsymbol{x}} \tilde{\boldsymbol{a}}^M$ as the vectorized version. We then define the standard architecture as $g_\phi(\vec{J}_{\boldsymbol{x}} \tilde{\boldsymbol{a}}^M)$, where $g_\phi : \mathbb{R}^{(M \times D)} \to \mathbb{R}^L$ is a fully-connected network.

We abbreviate the standard locally constant networks as **LCN**. Note that each $\boldsymbol{a}_1^i$ is locally linear and thus the Jacobian $J_{\boldsymbol{x}} \tilde{\boldsymbol{a}}^M$ is locally constant. We replace $\tilde{\boldsymbol{o}}^M$ with $J_{\boldsymbol{x}} \tilde{\boldsymbol{a}}^M$ as the invariant representation for each locally linear region[4], and replace the table $g$ with a differentiable function $g_\phi$ that takes as input real vectors. The gradients of LCN with respect to parameters is thus established through the derivatives of $g_\phi$ and the mixed partial derivatives of the neurons (derivatives of $\vec{J}_{\boldsymbol{x}} \tilde{\boldsymbol{a}}^M$).

Fortunately, all the previous analyses also apply to the standard architecture, due to a fine-grained equivalence between the two architectures.

**Theorem 8.** *Given any fixed $f_\theta$, any canonical locally constant network $g(\tilde{\boldsymbol{o}}^M)$ can be equivalently represented by a standard locally constant network $g_\phi(\vec{J}_{\boldsymbol{x}} \tilde{\boldsymbol{a}}^M)$, and vice versa.*

Since $f_\theta$ and $g$ control the decision nodes and leaf nodes in the associated oblique decision tree, respectively (see Theorem 3), Theorem 8 essentially states that both architectures are equally competent for assigning leaf nodes. Combining Theorem 8 with the analyses in §3.4, we have class-level equivalence among the two architectures of locally constant networks and oblique decision trees. The analyses in §3.5 are also inherited since the analyses only depend on decision nodes (i.e., $f_\theta$).

The core ideas for proving Theorem 8 are two-fold: 1) we find a bijection between the activation pattern $\tilde{\boldsymbol{o}}^M$ and the Jacobian $\vec{J}_{\boldsymbol{x}} \tilde{\boldsymbol{a}}^M$, and 2) feed-forward networks $g_\phi$ can map the (finitely many) Jacobian $\vec{J}_{\boldsymbol{x}} \tilde{\boldsymbol{a}}^M$ as flexibly as a table $g$. The complete proof is deferred to Appendix A.2.

**Discussion.** The standard architecture yields a new property that is only partially exhibited in the canonical architecture. For all the decision and leaf nodes which no training data is routed to, there is no way to obtain learning signals in classic oblique decision trees. However, due to shared parameterization (see §3.5), locally constant networks can "learn" all the decision nodes in the implied oblique decision trees (if there is a way to optimize the networks), and the standard architecture can even "learn" all the leaf nodes due to the parameterized output function $g_\phi$.

**Extensions.** The construction of (standard) locally constant networks enables several natural extensions due to the flexibility of the neural architecture and the interpretation of decision trees. The original locally linear networks (**LLN**) $f_\theta$, which outputs a linear function instead of a constant function for each region, can be regarded as one extension. Here we discuss two examples.

- Approximately locally constant networks (**ALCN**): we can change the activation function while keeping the model architecture of LCN. For example, we can replace ReLU $\max(0, x)$ with softplus $\log(1 + \exp(x))$, which will lead to an approximately locally constant network, as the softplus function has an approximately locally constant derivative for inputs with large absolute value. Note that the canonical architecture (tabular $g$) is not compatible with such extension.

---

[4]In practice, we also include each bias $\boldsymbol{a}_1^i - (\nabla_{\boldsymbol{x}} \boldsymbol{a}_1^i)^\top \boldsymbol{x}$, which is omitted here to simplify exposition.

- Ensemble locally constant networks (**ELCN**): since each LCN can only output $2^M$ different values, it is limited for complex tasks like regression (akin to decision trees). We can instead use an additive ensemble of LCN or ALCN to increase the capacity. We use $g_\phi^{[e]}(\vec{J}_{\boldsymbol{x}} \tilde{\boldsymbol{a}}^{M,[e]})$ to denote a base model in the ensemble, and denote the ensemble with $E$ models as $\sum_{e=1}^{E} g_\phi^{[e]}(\vec{J}_{\boldsymbol{x}} \tilde{\boldsymbol{a}}^{M,[e]})$.

### 3.7 COMPUTATION AND LEARNING

In this section, we discuss computation and learning algorithms for the proposed models. In the following complexity analyses, we assume $g_\phi$ to be a linear model.

**Space complexity.** The space complexity of LCN is $\Theta(MD)$ for representing decision nodes and $\Theta(MDL)$ for representing leaf nodes. In contrast, the space complexity of classic oblique decision trees is $\Theta((2^M - 1)D)$ for decision nodes and $\Theta(2^M L)$ for leaf nodes. Hence, our representation improves the space complexity over classic oblique decision trees exponentially.

**Computation and time complexity.** LCN and ALCN are built on the gradients of all the neurons $\vec{J}_{\boldsymbol{x}} \tilde{\boldsymbol{a}}^M = [\nabla_{\boldsymbol{x}} \boldsymbol{a}_1^M, \ldots, \nabla_{\boldsymbol{x}} \boldsymbol{a}_1^1]$, which can be computationally challenging to obtain. Existing automatic differentiation (e.g., back-propagation) only computes the gradient of a scalar output. Instead, here we propose an efficient dynamic programming procedure which only requires a forward pass:

1. $\nabla_{\boldsymbol{x}} \boldsymbol{a}_1^1 = \boldsymbol{o}_1^1 \times \boldsymbol{W}^1$.
2. $\forall i \in \{2, \ldots, M\}, \nabla_{\boldsymbol{x}} \boldsymbol{a}_1^i = \boldsymbol{o}_1^i \times (\boldsymbol{W}_{1,1:D}^i + \sum_{k=1}^{i-1} \boldsymbol{W}_{1,D+k}^i \nabla_{\boldsymbol{x}} \boldsymbol{a}_1^k)$,

The complexity of the dynamic programming is $\Theta(M^2)$ due to the inner-summation inside each iteration. Straightforward back-propagation re-computes the partial solutions $\nabla_{\boldsymbol{x}} \boldsymbol{a}_1^k$ for each $\nabla_{\boldsymbol{x}} \boldsymbol{a}_1^i$, so the complexity is $\Theta(M^3)$. We can parallelize the inner-summation on a GPU, and the complexity of the dynamic programming and straightforward back-propagation will become $\Theta(M)$ and $\Theta(M^2)$, respectively. Note that the complexity of a forward pass of a typical network is also $\Theta(M)$ on a GPU. The time complexity of learning LCN by (stochastic) gradient descent is thus $\Theta(M\tau)$, where $\tau$ denotes the number of iterations. In contrast, the computation of existing oblique decision tree training algorithms is typically data-dependent and thus the complexity is hard to characterize.

**Training LCN and ALCN.** Even though LCN is sub-differentiable, whenever $\boldsymbol{o}_1^i = 0$, the network does not exhibit useful gradient information for learning each locally constant representation $\nabla_{\boldsymbol{x}} \boldsymbol{a}_1^i$ (note that $\vec{J}_{\boldsymbol{x}} \tilde{\boldsymbol{a}}^M = [\nabla_{\boldsymbol{x}} \boldsymbol{a}_1^1, \ldots, \nabla_{\boldsymbol{x}} \boldsymbol{a}_1^M]$), since, operationally, $\boldsymbol{o}_1^i = 0$ implies $\boldsymbol{a}_1^i \leftarrow 0$ and there is no useful gradient of $\nabla_{\boldsymbol{x}} \boldsymbol{a}_1^i = \nabla_{\boldsymbol{x}} 0 = \boldsymbol{0}$ with respect to model parameters. To alleviate the problem, we propose to leverage softplus as an infinitely differentiable approximation of ReLU to obtain meaningful learning signals for $\nabla_{\boldsymbol{x}} \boldsymbol{a}_1^i$. Concretely, we conduct the annealing during training:

$$\boldsymbol{a}_1^i = \lambda_t \max(0, \boldsymbol{z}_1^i) + (1 - \lambda_t) \log(1 + \exp(\boldsymbol{z}_1^i)), \forall i \in [M], \lambda_t \in [0, 1], \quad (5)$$

where $\lambda_t$ is an iteration-dependent annealing parameter. Both LCN and ALCN can be constructed as a special case of Eq. (5). We train LCN with $\lambda_t$ equal to the ratio between the current epoch and the total epochs, and ALCN with $\lambda_t = 0$. Both models are optimized via stochastic gradient descent.

We also include DropConnect (Wan et al., 2013) to the weight matrices $\boldsymbol{W}^i \leftarrow \text{drop}(\boldsymbol{W}^i)$ during training. Despite the simple structure of DropConnect in the locally constant networks, it entails a structural dropout on the weights in the corresponding oblique decision trees (see §3.5), which is challenging to reproduce in typical oblique decision trees. In addition, it also encourages the exploration of parameter space, which is easy to see for the raw LCN: the randomization enables the exploration that flips $\boldsymbol{o}_1^i = 0$ to $\boldsymbol{o}_1^i = 1$ to establish effective learning signal. Note that the standard DropOut (Srivastava et al., 2014) is not ideal for the low capacity models that we consider here.

**Training ELCN.** Since each ensemble component is sub-differentiable, we can directly learn the whole ensemble through gradient descent. However, the approach is not scalable due to memory constraints in practice. Instead, we propose to train the ensemble in a boosting fashion:

1. We first train an initial locally constant network $g_\phi^{[1]}(\vec{J}_{\boldsymbol{x}} \tilde{\boldsymbol{a}}^{M,[1]})$.
2. For each iteration $e' \in \{2, 3, \ldots, E\}$, we incrementally optimize $\sum_{e=1}^{e'} g_\phi^{[e]}(\vec{J}_{\boldsymbol{x}} \tilde{\boldsymbol{a}}^{M,[e]})$.

Note that, in the second step, only the latest model is optimized, and thus we can simply store the predictions of the preceding models without loading them into the memory. Each partial ensemble can be directly learned through gradient descent, without resorting to complex meta-algorithms such as adaptive boosting (Freund & Schapire, 1997) or gradient boosting (Friedman, 2001).

Table 1: Dataset statistics

| Dataset | Bace | HIV | SIDER | Tox21 | PDBbind |
|---|---|---|---|---|---|
| Task | (Multi-label) binary classification | | | | Regression |
| Number of labels | 1 | 1 | 27 | 12 | 1 |
| Number of data | 1,513 | 41,127 | 1,427 | 7,831 | 11,908 |

Table 2: Main results. The $1^{st}$ section refers to (oblique) decision tree methods, the $2^{nd}$ section refers to single model extensions of LCN, the $3^{rd}$ section refers to ensemble methods, and the last section is GCN. The results of GCN are copied from (Wu et al., 2018), where the results in SIDER and Tox21 are not directly comparable due to lack of standard splittings. The best result in each section is in bold letters.

| Dataset | Bace (AUC) | HIV (AUC) | SIDER (AUC) | Tox21 (AUC) | PDBbind (RMSE) |
|---|---|---|---|---|---|
| CART | $0.652 \pm 0.024$ | $0.544 \pm 0.009$ | $0.570 \pm 0.010$ | $0.651 \pm 0.005$ | $1.573 \pm 0.000$ |
| HHCART | $0.545 \pm 0.016$ | $0.636 \pm 0.000$ | $0.570 \pm 0.009$ | $0.638 \pm 0.007$ | $1.530 \pm 0.000$ |
| TAO | $0.734 \pm 0.000$ | $0.627 \pm 0.000$ | $0.577 \pm 0.004$ | $0.676 \pm 0.003$ | Not applicable |
| LCN | $\mathbf{0.839 \pm 0.013}$ | $\mathbf{0.728 \pm 0.013}$ | $\mathbf{0.624 \pm 0.044}$ | $\mathbf{0.781 \pm 0.017}$ | $\mathbf{1.508 \pm 0.017}$ |
| LLN | $0.818 \pm 0.007$ | $0.737 \pm 0.009$ | $\mathbf{0.677 \pm 0.014}$ | $0.813 \pm 0.009$ | $1.627 \pm 0.008$ |
| ALCN | $\mathbf{0.854 \pm 0.007}$ | $\mathbf{0.738 \pm 0.009}$ | $0.653 \pm 0.044$ | $\mathbf{0.814 \pm 0.009}$ | $\mathbf{1.369 \pm 0.007}$ |
| RF | $0.869 \pm 0.003$ | $\mathbf{0.796 \pm 0.007}$ | $0.685 \pm 0.011$ | $\mathbf{0.839 \pm 0.007}$ | $1.256 \pm 0.002$ |
| GBDT | $0.859 \pm 0.005$ | $0.748 \pm 0.001$ | $0.668 \pm 0.014$ | $0.812 \pm 0.011$ | $1.247 \pm 0.002$ |
| ELCN | $\mathbf{0.874 \pm 0.005}$ | $0.757 \pm 0.011$ | $\mathbf{0.685 \pm 0.010}$ | $0.822 \pm 0.006$ | $\mathbf{1.219 \pm 0.007}$ |
| GCN | $0.783 \pm 0.014$ | $0.763 \pm 0.016$ | $*0.638 \pm 0.012$ | $*0.829 \pm 0.006$ | $1.44 \pm 0.12$ |

## 4 EXPERIMENT

Here we evaluate the efficacy of our models (LCN, ALCN, and ELCN) using the chemical property prediction datasets from MoleculeNet (Wu et al., 2018), where random forest performs competitively. We include 4 (multi-label) binary classification datasets and 1 regression dataset. The statistics are available in Table 1. We follow the literature to construct the feature (Wu et al., 2018). Specifically, we use the standard Morgan fingerprint (Rogers & Hahn, 2010), 2,048 binary indicators of chemical substructures, for the classification datasets, and 'grid features' (fingerprints of pairs between ligand and protein, see Wu et al. (2018)) for the regression dataset. Each dataset is splitted into (train, validation, test) sets under the criterion specified in MoleculeNet.

We compare LCN and its extensions (LLN, ALCN, and ELCN) with the following baselines:

- (Oblique) decision trees: CART (Breiman et al. (1984)), HHCART (Wickramarachchi et al. (2016); oblique decision trees induced greedily on linear projections), and TAO (Carreira-Perpiñán & Tavallali (2018); oblique decision trees trained via alternating optimization).
- Tree ensembles: RF (Breiman (2001); random forest) and GBDT (Friedman (2001); gradient boosting decision trees).
- Graph networks: GCN (Duvenaud et al. (2015); graph convolutional networks on molecules).

For decision trees, LCN, LLN, and ALCN, we tune the tree depth in $\{2, 3, \ldots, 12\}$. For LCN, LLN, and ALCN, we also tune the DropConnect probability in $\{0, 0.25, 0.5, 0.75\}$. Since regression tasks require precise estimations of the prediction values while classification tasks do not, we tune the number of hidden layers of $g_\phi$ in $\{0, 1, 2, 3, 4\}$ (each with 256 neurons) for the regression task, and simply use a linear model $g_\phi$ for the classification tasks. For ELCN, we use ALCN as the base model, tune the ensemble size $E \in \{2^0, 2^1, \ldots, 2^6\}$ for the classification tasks, and $E \in \{2^0, 2^1, \ldots, 2^9\}$ for the regression task. To train our models, we use the cross entropy loss for the classification tasks, and mean squared error for the regression task. Other minor details are available in Appendix B.

We follow the chemistry literature (Wu et al., 2018) to measure the performance by AUC for classification, and root-mean-squared error (RMSE) for regression. For each dataset, we train a model for

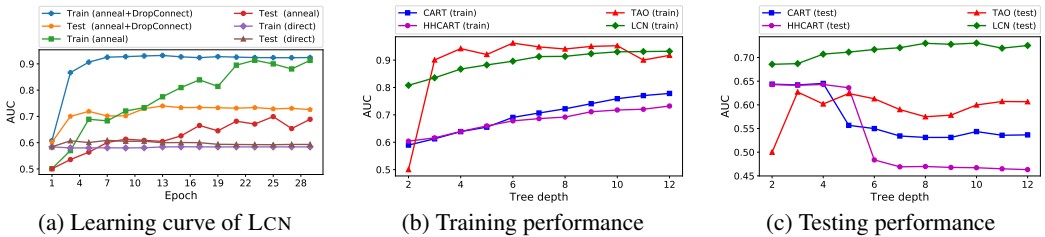

Figure 2: Empirical analysis for oblique decision trees on the HIV dataset. Fig. 2a is an ablation study for LCN and Fig. 2b-2c compare different training methods.

each label, compute the mean and standard deviation of the performance across 10 different random seeds, and report their average across all the labels within the dataset. The results are in Table 2.

Among the (oblique) decision tree training algorithms, our LCN achieves the state-of-the-art performance. The continuous extension (ALCN) always improves the empirical performance of LCN, which is expected since LCN is limited for the number of possible outputs (leaf nodes). Among the ensemble methods, the proposed ELCN always outperforms the classic counterpart, GBDT, and sometimes outperforms RF. Overall, LCN is the state-of-the-art method for learning oblique decision trees, and ELCN performs competitively against other alternatives for training tree ensembles.

**Empirical analysis.** Here we analyze the proposed LCN in terms of the optimization and generalization performance in the large HIV dataset. We conduct an ablation study on the proposed method for training LCN in Figure 2a. Direct training (without annealing) does not suffice to learn LCN, while the proposed annealing succeed in optimization; even better optimization and generalization performance can be achieved by introducing DropConnect, which corroborates our hypothesis on the exploration effect during training in §3.7 and its well-known regularization effect. Compared to other methods (Fig. 2b), only TAO has a comparable training performance. In terms of generalization (Fig. 2c), all of the competitors do not perform well and overfit fairly quickly. In stark contrast, LCN outperforms the competitors by a large margin and gets even more accurate as the depth increases. This is expected due to the strong regularization of LCN that uses a linear number of effective weights to construct an exponential number of decision nodes, as discussed in §3.5. Some additional analysis and the visualization of the tree converted from LCN are included in Appendix C.

## 5 DISCUSSION AND CONCLUSION

We create a novel neural architecture by casting the derivatives of deep networks as the representation, which realizes a new class of neural models that is equivalent to oblique decision trees. The induced oblique decision trees embed rich structures and are compatible with deep learning methods. This work can be used to interpret methods that utilize derivatives of a network, such as training a generator through the gradient of a discriminator (Goodfellow et al., 2014). The work opens up many avenues for future work, from building representations from the derivatives of neural models to the incorporation of more structures, such as the inner randomization of random forest.

ACKNOWLEDGMENTS

GH and TJ were in part supported by a grant from Siemens Corporation. The authors thank Shubhendu Trivedi and Menghua Wu for proofreading, and thank the anonymous reviewers for their helpful comments.

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

# A    PROOFS

## A.1    PROOF OF LEMMA 6

*Proof.* We fix $j$ and do induction on $i$. Without loss of generality, we assume $1 = \boldsymbol{r}_j \neq \boldsymbol{r}'_j = 0$.

If $i = j$, since $\boldsymbol{r}'_j = 0$, we have

$$\begin{cases} \boldsymbol{\omega}_{\boldsymbol{r}_{1:i}} = \boldsymbol{W}^{i+1}_{1,1:D} + \sum_{k=1}^{i} \boldsymbol{W}^{i+1}_{1,D+k} \times \boldsymbol{r}_k \times \boldsymbol{\omega}_{\boldsymbol{r}_{1:k-1}}, \\ \boldsymbol{\omega}_{\boldsymbol{r}'_{1:1}} = \boldsymbol{W}^{i+1}_{1,1:D} + \sum_{k=1}^{i-1} \boldsymbol{W}^{i+1}_{1,D+k} \times \boldsymbol{r}_k \times \boldsymbol{\omega}_{\boldsymbol{r}_{1:k-1}}. \end{cases}$$

Hence, we have $\boldsymbol{\omega}_{\boldsymbol{r}_{1:i}} - \boldsymbol{\omega}_{\boldsymbol{r}'_{1:1}} = (\boldsymbol{W}^{i+1}_{1,D+i} \times \boldsymbol{r}_i) \times \boldsymbol{\omega}_{\boldsymbol{r}_{1:i-1}} = \alpha \times \boldsymbol{\omega}_{\boldsymbol{r}_{1:j-1}}$.

We assume the statement holds for *up to* some integer $i \geq j$:

$$\boldsymbol{\omega}_{\boldsymbol{r}_{1:i}} - \boldsymbol{\omega}_{\boldsymbol{r}'_{1:i}} = \alpha \times \boldsymbol{\omega}_{\boldsymbol{r}_{1:j-1}}, \text{ for some } \alpha \in \mathbb{R}.$$

For $i + 1$, we have

$$\begin{aligned} \boldsymbol{\omega}_{\boldsymbol{r}_{1:i+1}} =& \boldsymbol{W}^{i+2}_{1,1:D} + \sum_{k=1}^{i+1} \boldsymbol{W}^{i+2}_{1,D+k} \times \boldsymbol{r}_k \times \boldsymbol{\omega}_{\boldsymbol{r}_{1:k-1}} \\ =& \boldsymbol{W}^{i+2}_{1,1:D} + \sum_{k=1}^{j-1} \boldsymbol{W}^{i+2}_{1,D+k} \times \boldsymbol{r}_k \times \boldsymbol{\omega}_{\boldsymbol{r}_{1:k-1}} + \boldsymbol{W}^{i+2}_{1,D+j} \times \boldsymbol{r}_j \times \boldsymbol{\omega}_{\boldsymbol{r}_{1:j-1}} \\ & + \sum_{k=j+1}^{i+1} \boldsymbol{W}^{i+2}_{1,D+k} \times \boldsymbol{r}_k \times \boldsymbol{\omega}_{\boldsymbol{r}_{1:k-1}} \\ =& \boldsymbol{W}^{i+2}_{1,1:D} + \sum_{k=1}^{j-1} \boldsymbol{W}^{i+2}_{1,D+k} \times \boldsymbol{r}'_k \times \boldsymbol{\omega}_{\boldsymbol{r}'_{1:k-1}} + \boldsymbol{W}^{i+2}_{1,D+j} \times \boldsymbol{r}_j \times \boldsymbol{\omega}_{\boldsymbol{r}_{1:j-1}} \\ & + \sum_{k=j+1}^{i+1} \boldsymbol{W}^{i+2}_{1,D+k} \times \boldsymbol{r}'_k \times (\boldsymbol{\omega}_{\boldsymbol{r}'_{1:k-1}} + \alpha_k \times \boldsymbol{\omega}_{\boldsymbol{r}_{1:j-1}}), \text{ for some } \alpha_k \in \mathbb{R} \\ =& \boldsymbol{W}^{i+2}_{1,1:D} + \sum_{k=1}^{i+1} \boldsymbol{W}^{i+2}_{1,D+k} \times \boldsymbol{r}'_k \times \boldsymbol{\omega}_{\boldsymbol{r}'_{1:k-1}} \\ & + (\boldsymbol{W}^{i+2}_{1,D+j} \times \boldsymbol{r}_j + \sum_{k=j+1}^{i+1} \boldsymbol{W}^{i+2}_{1,D+k} \times \boldsymbol{r}_k \times \alpha_k) \times \boldsymbol{\omega}_{\boldsymbol{r}_{1:j-1}} \\ =& \boldsymbol{\omega}_{\boldsymbol{r}'_{1:i+1}} + \alpha \times \boldsymbol{\omega}_{\boldsymbol{r}_{1:j-1}}, \text{ for some } \alpha \in \mathbb{R} \end{aligned}$$

The proof follows by induction.    □

## A.2    PROOF OF THEOREM 8

*Proof.* We first prove that we can represent any $g_\phi(\vec{J}_{\boldsymbol{x}} \tilde{\boldsymbol{a}}^M)$ as $g(\tilde{\boldsymbol{o}}^M)$. Note that for any $\boldsymbol{x}$ mapping to the same activation pattern $\tilde{\boldsymbol{o}}^M$, the Jacobian $\vec{J}_{\boldsymbol{x}} \tilde{\boldsymbol{a}}^M$ is constant. Hence, we may re-write the standard architecture $g_\phi(\vec{J}_{\boldsymbol{x}} \tilde{\boldsymbol{a}}^M)$ as $g_\phi(\vec{J}(\tilde{\boldsymbol{o}}^M))$, where $\vec{J}(\tilde{\boldsymbol{o}}^M)$ is the Jacobian corresponding to the activation pattern $\tilde{\boldsymbol{o}}^M$. Then we can set $g(\cdot) \triangleq g_\phi(\vec{J}(\cdot))$, which concludes the first part of the proof.

To prove the other direction, we first prove that we can also write the activation pattern as a function of the Jacobian. We prove this by layer-wise induction (note that $\tilde{\boldsymbol{o}}^M = [\boldsymbol{o}_1^1, \dots, \boldsymbol{o}_1^M]$ and $\vec{J}_{\boldsymbol{x}} \tilde{\boldsymbol{a}}^M = [\nabla_{\boldsymbol{x}} \boldsymbol{a}_1^1, \dots, \nabla_{\boldsymbol{x}} \boldsymbol{a}_1^M]$):

1. The induction hypothesis ($i \geq 2$) is that $[\boldsymbol{o}_1^1, \dots, \boldsymbol{o}_1^{i-1}]$ is a function of $[\nabla_{\boldsymbol{x}} \boldsymbol{a}_1^1, \dots, \nabla_{\boldsymbol{x}} \boldsymbol{a}_1^{i-1}]$.

2. If $\boldsymbol{W}^1 = \boldsymbol{0}$ (zero vector), $\boldsymbol{z}_1^1$, $\boldsymbol{a}_1^1$, and $\boldsymbol{o}_1^1$ are constant (thus being a function of $\nabla_{\boldsymbol{x}} \boldsymbol{a}_1^1$). Otherwise, $\nabla_{\boldsymbol{x}} \boldsymbol{a}_1^1 = \boldsymbol{0} \Leftrightarrow \boldsymbol{o}_1^1 = 0$ and $\nabla_{\boldsymbol{x}} \boldsymbol{a}_1^1 = \boldsymbol{W}^1 \Leftrightarrow \boldsymbol{o}_1^1 = 1$, so $\boldsymbol{o}_1^1$ can be written as a function of $\nabla_{\boldsymbol{x}} \boldsymbol{a}_1^1$.

3. Assume that we are given $[\nabla_{\boldsymbol{x}}\boldsymbol{a}_1^1, \ldots, \nabla_{\boldsymbol{x}}\boldsymbol{a}_1^{i-1}]$ and the corresponding $[\boldsymbol{o}_1^1, \ldots, \boldsymbol{o}_1^{i-1}]$.

If either $\boldsymbol{o}_1^i = 0$ or $\boldsymbol{o}_1^i = 1$ is infeasible (but not both), by induction hypothesis, $[\boldsymbol{o}_1^1, \ldots, \boldsymbol{o}_1^i]$ can be written as a function of $[\nabla_{\boldsymbol{x}}\boldsymbol{a}_1^1, \ldots, \nabla_{\boldsymbol{x}}\boldsymbol{a}_1^i]$.

If $\boldsymbol{o}_1^i = 1$ for some $\boldsymbol{x}'$ and $\boldsymbol{o}_1^i = 0$ for some $\boldsymbol{x}''$, we claim that $\boldsymbol{o}_1^i = 1 \Rightarrow \nabla_{\boldsymbol{x}}\boldsymbol{a}_1^i \neq \boldsymbol{0}$:

If $\boldsymbol{o}_1^i = 1$ and $\nabla_{\boldsymbol{x}}\boldsymbol{a}_1^i = \boldsymbol{0}$, we have $\boldsymbol{o}_1^i = 1 \Rightarrow \boldsymbol{a}_1^i = \boldsymbol{z}_1^i \geq 0$ and $\boldsymbol{0} = \nabla_{\boldsymbol{x}}\boldsymbol{a}_1^i = \nabla_{\boldsymbol{x}}\boldsymbol{z}_1^i$, which implies that the bias (of $\boldsymbol{z}^i$, given $[\boldsymbol{o}_1^1, \ldots, \boldsymbol{o}_1^{i-1}]$) $\boldsymbol{z}_1^i - (\nabla_{\boldsymbol{x}}\boldsymbol{z}_1^i)^\top \boldsymbol{x} \geq 0$. Note that both $\nabla_{\boldsymbol{x}}\boldsymbol{z}_1^i$ and $\boldsymbol{z}_1^i - (\nabla_{\boldsymbol{x}}\boldsymbol{z}_1^i)^\top \boldsymbol{x}$ are constant given $[\boldsymbol{o}_1^1, \ldots, \boldsymbol{o}_1^{i-1}]$, regardless of $\boldsymbol{o}_1^i$. Hence, given $[\boldsymbol{o}_1^1, \ldots, \boldsymbol{o}_1^{i-1}]$, we have $\boldsymbol{z}_1^i = \boldsymbol{z}_1^i - (\nabla_{\boldsymbol{x}}\boldsymbol{z}_1^i)^\top \boldsymbol{x} \geq 0$ and $\boldsymbol{o}_1^i = 0$ is infeasible ($\Rightarrow\!\!\Leftarrow$).

Note that, given fixed $[\boldsymbol{o}_1^1, \ldots, \boldsymbol{o}_1^{i-1}]$, $\nabla_{\boldsymbol{x}}\boldsymbol{a}_1^i \neq \boldsymbol{0}$ has a unique value in $\mathbb{R}^d$. Combining the result $\boldsymbol{o}_1^i = 1 \Rightarrow \nabla_{\boldsymbol{x}}\boldsymbol{a}_1^i \neq \boldsymbol{0}$ with $\boldsymbol{o}_1^i = 0 \Rightarrow \nabla_{\boldsymbol{x}}\boldsymbol{a}_1^i = \boldsymbol{0}$, there is a bijection between $\boldsymbol{o}_1^i$ and $\nabla_{\boldsymbol{x}}\boldsymbol{a}_1^i$ in this case, which implies that $[\boldsymbol{o}_1^1, \ldots, \boldsymbol{o}_1^i]$ can be written as a function of $[\nabla_{\boldsymbol{x}}\boldsymbol{a}_1^1, \ldots, \nabla_{\boldsymbol{x}}\boldsymbol{a}_1^i]$.

The derivation implies that we may re-write the canonical architecture $g(\tilde{\boldsymbol{o}}^M)$ as $g(\tilde{\boldsymbol{o}}(\vec{J}_{\boldsymbol{x}}\tilde{\boldsymbol{a}}^M))$, where $\tilde{\boldsymbol{o}}(\vec{J}_{\boldsymbol{x}}\tilde{\boldsymbol{a}}^M)$ is the activation pattern corresponding to the Jacobian $\vec{J}_{\boldsymbol{x}}\tilde{\boldsymbol{a}}^M$. Hence, it suffices to establish that there exists a feed-forward network $g_\phi$ such that $g_\phi(\vec{J}_{\boldsymbol{x}}\tilde{\boldsymbol{a}}^M) = g(\tilde{\boldsymbol{o}}(\vec{J}_{\boldsymbol{x}}\tilde{\boldsymbol{a}}^M))$ for at most $2^M$ distinct $\vec{J}_{\boldsymbol{x}}\tilde{\boldsymbol{a}}^M$, which can be found by the Theorem 2.5 of Hornik et al. (1989) or the Theorem 1 of Zhang et al. (2017). $\square$

# B IMPLEMENTATION DETAILS

Here we provide the full version of the implementation details.

For the baseline methods:

- CART, HHCART, and TAO: we tune the tree depth in $\{2, 3, \ldots, 12\}$.
- RF: we use the `scikit-learn` (Pedregosa et al., 2011) implementation of random forest. We set the number of estimators as $500$.
- GBDT: we use the `scikit-learn` (Pedregosa et al., 2011) implementation of gradient boosting trees. We tune the number of estimators in $\{2^3, 2^4, \ldots, 2^{10}\}$.

For LCN, LLN, and ALCN, we run the same training procedure. For all the datasets, we tune the depth in $\{2, 3, \ldots, 12\}$ and the DropConnect probability in $\{0, 0.25, 0.5, 0.75\}$. The models are optimized with mini-batch stochastic gradient descent with batch size set to $64$. For all the classification tasks, we set the learning rate as $0.1$, which is annealed by a factor of $10$ for every $10$ epochs ($30$ epochs in total). For the regression task, we set the learning rate as $0.0001$, which is annealed by a factor of $10$ for every $30$ epochs ($60$ epochs in total).

Both LCN and ALCN have an extra fully-connected network $g_\phi$, which transforms the derivatives $\vec{J}_{\boldsymbol{x}}\tilde{\boldsymbol{a}}^M$ to the final outputs. Since regression tasks require precise estimation of prediction values while classification tasks do not, we tune the number of hidden layers of $g_\phi$ in $\{0, 1, 2, 3, 4\}$ (each with $256$ neurons) for the regression dataset, and simply use a linear $g_\phi$ for the classification datasets.

For ELCN, we fix the depth to $12$ and tune the number of base models $E \in \{2^0, 2^1, \ldots, 2^6\}$ for the classification tasks, and $E \in \{2^0, 2^1, \ldots, 2^9\}$ for the regression task. We set the DropConnect probability as $0.75$ to encourage strong regularization for the classification tasks, and as $0.25$ to impose mild regularization for the regression task (because regression is hard to fit). We found stochastic gradient descent does not suffice to incrementally learn the ELCN, so we use the AMSGrad optimizer (Reddi et al., 2018) instead. We set the batch size as $256$ and train each partial ensemble for $30$ epochs. The learning rate is $0.01$ for the classification tasks, and $0.0001$ for the regression task.

To train our models, we use the cross entropy loss for the classification tasks, and mean squared error for the regression task.

Table 3: Analysis for "unobserved decision patterns" of LCN in the Bace dataset.

| Depth | 8 | 9 | 10 | 11 | 12 |
|---|---|---|---|---|---|
| # of possible patterns | 256 | 512 | 1024 | 2048 | 4096 |
| # of training patterns | 72 | 58 | 85 | 103 | 86 |
| # of testing patterns | 32 | 31 | 48 | 49 | 40 |
| # of testing patterns - training patterns | 5 | 2 | 11 | 8 | 11 |
| Ratio of testing points w/ unobserved patterns | 0.040 | 0.013 | 0.072 | 0.059 | 0.079 |
| Testing performance - observed patterns | 0.8505 | 0.8184 | 0.8270 | 0.8429 | 0.8390 |
| Testing performance - unobserved patterns | 0.8596 | 0.9145 | 0.8303 | 0.7732 | 0.8894 |

## C SUPPLEMENTARY EMPIRICAL ANALYSIS AND VISUALIZATION

### C.1 SUPPLEMENTARY EMPIRICAL ANALYSIS

In this section, we investigate the learning of "unobserved branching / leaves" discussed in §3.6. The "unobserved branching / leaves" refer to the decision and leaf nodes of the oblique decision tree converted from LCN, such that there is no training data that are routed to the nodes. It is impossible for traditional (oblique) decision tree training algorithms to learn the values of such nodes (e.g., the output value of a leaf node in the traditional framework is based on the training data that are routed to the leaf node). However, the shared parameterization in our oblique decision tree provides a means to update such unobserved nodes during training (see the discussion in §3.6).

Since the above scenario in general happens more frequently in small datasets than in large datasets, we evaluate the scenario on the small Bace dataset (binary classification task). Here we empirically analyze a few things pertaining to the unobserved nodes:

- # of training patterns: the number of distinct end-to-end activation / decision patterns $r_{1:M}$ encountered in the training data.
- # of testing patterns: the number of distinct end-to-end activation / decision patterns $r_{1:M}$ encountered ib the testing data.
- # of testing patterns - training patterns: the number of distinct end-to-end activation / decision patterns $r_{1:M}$ that is only encountered in the testing data but not in the training data.
- Ratio of testing points w/ unobserved patterns: the number of testing points that yield unobserved patterns divided by the total number of testing points.
- Testing performance - observed patterns: here we denote the number of testing data as $n$, the prediction and label of the $i^{\text{th}}$ as $\hat{y}_i \in [0, 1]$ and $y_i \in \{0, 1\}$, respectively. We collect the subset of indices $I$ of the testing data such that their activation / decision patterns $r_{1:M}$ are observed in the training data, and then compute the performance of their predictions. Since the original performance is measured by AUC, here we generalize AUC to measure a subset of points $I$ as:

$$\frac{\sum_{i \in I} \sum_{j=1}^{n} \left( \mathbb{I}[y_i > y_j]\left( \mathbb{I}[\hat{y}_i > \hat{y}_j] + 0.5\mathbb{I}[\hat{y}_i = \hat{y}_j] \right) + \mathbb{I}[y_i < y_j]\left( \mathbb{I}[\hat{y}_i < \hat{y}_j] + 0.5\mathbb{I}[\hat{y}_i = \hat{y}_j] \right) \right)}{\sum_{i \in I} \sum_{j=1}^{n} \left( \mathbb{I}[y_i > y_j] + \mathbb{I}[y_i < y_j]) \right)}. \tag{6}$$

  When $I = [n]$, the above measure recovers AUC.
- Testing performance - unobserved patterns: the same as above, but use $I$ for the testing data such that their activation / decision patterns $r_{1:M}$ are *unobserved* in the training data.

The results are in Table 3. There are some interesting findings. For example, there is an exponential number of possible patterns, but the number of patterns that appear in the dataset is quite small. The ratio of testing points with unobserved patterns is also small, but these unobserved branching / leaves seem to be controlled properly. They do not lead to completely different performance compared to those that are observed during training.

### C.2 VISUALIZATION

Here we visualize the learned locally constant network on the HIV dataset in the representation of its equivalent oblique decision tree in Fig. 3. Since the dimension of Morgan fingerprint (Rogers &

Hahn, 2010) is quite high (2,048), here we only visualize the top-K weights (in terms of the absolute value) for each decision node. We also normalize each weight such that the $\ell_1$ norm of each weight is 1. Since the task is evaluated by ranking (AUC), we visualize the leaf nodes in terms of the ranking of output probability among the leaf nodes (the higher the more likely).

Note that a complete visualization requires some engineering efforts. Our main contribution here is the algorithm that transforms an LCN to an oblique decision tree, rather than the visualization of oblique decision trees, so we only provide the initial visualization as a proof of concept.

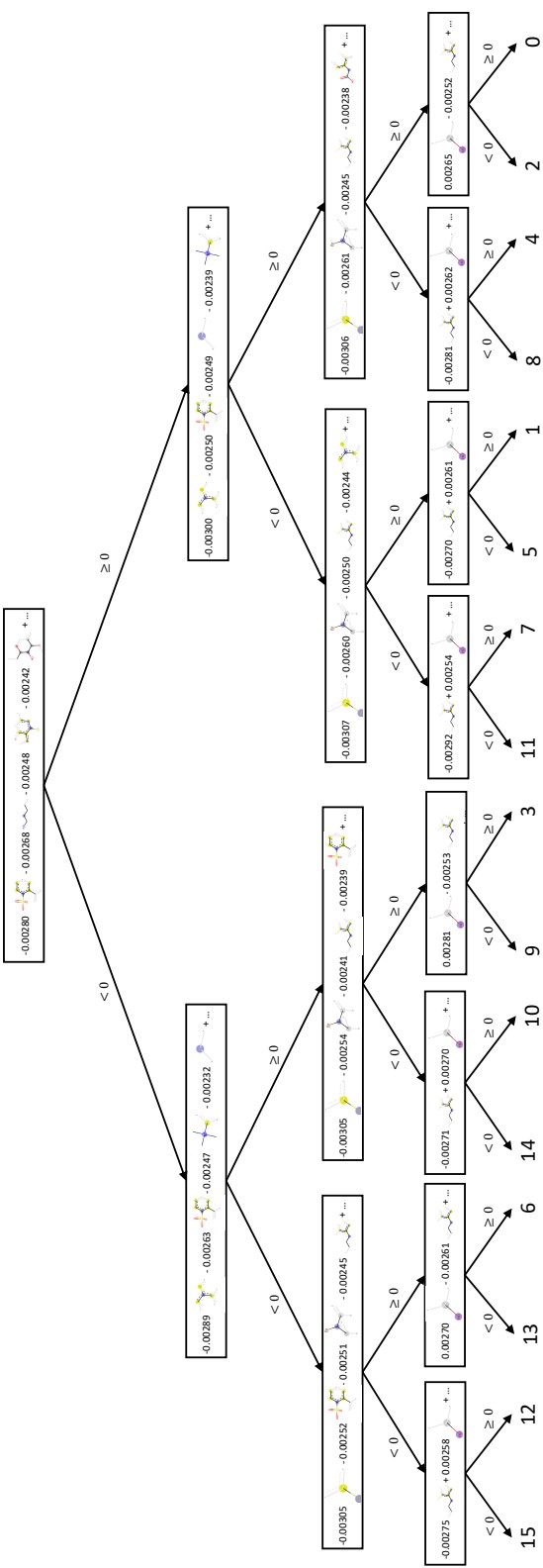

Figure 3: Visualization of learned locally constant network in the representation of oblique decision trees using the proof of Theorem 3. The number in the leaves indicates the ranking of output probability among the 16 leaves (the exact value is not important). See the descriptions in Appendix C.2.

