# OpenReview forum: "Oblique Decision Trees from Derivatives of ReLU Networks"
_ICLR.cc/2020/Conference — Accept (Poster)_

### Official Review · AnonReviewer2 · 2019-10-22
**Official Blind Review #2**

**Rating:** 6

**Review:**

In this paper, the authors proposed an approach to fit locally constant functions using deep neural networks (DNNs).
The idea is based on the fact that DNN consisting of only linear transformations and ReLU activations is piecewise linear.
Thus, the derivative of such a network with respect to the input is locally constant.

In the paper, the authors focused on connecting the locally constant network with oblique decision trees.
Specifically, they proved that these two models are in some sense equivalent, and one can transform one model to another.
This connection enables us to train the oblique decision trees by training the locally constant network instead.
Because the locally constant network can be trained using the gradient-based methods, it would be much easier to train than the oblique decision trees.

I think the paper is well-written and the idea is clear.
Connecting the locally constant network with oblique decision trees looks interesting.

I have one concern, however.
The authors mention that the training of oblique decision trees is difficult, and the use of the locally constant network is helpful.
If I understand correctly, oblique decision tree is one specific instance of the hierarchical mixtures of experts.
And, [Ref1] pointed out that the hierarchical mixtures of experts can be trained using EM algorithm, which is another type of the gradient-based training.
The current paper misses such a prior study.
I am interested in to see if the use of locally constant network is truly effective for training oblique decision trees over the algorithms considered in the literatures of hierarchical mixtures of experts.

[Ref1] Hierarchical mixtures of experts and the EM algorithm


### Updated after author response ###
The authors have successfully demonstrated that the proposed approach is better than the EM-like classical approaches. I therefore keep my score.

**Experience Assessment:**

I have read many papers in this area.

**Review Assessment: Checking Correctness Of Derivations And Theory:**

I did not assess the derivations or theory.

**Review Assessment: Checking Correctness Of Experiments:**

I assessed the sensibility of the experiments.

**Review Assessment: Thoroughness In Paper Reading:**

I made a quick assessment of this paper.

---

> ### Author Response · Authors · 2019-11-10
> **Official response**
>
> We thank the reviewer for the insightful comments and suggestions.
>
> To clarify the concern, there is a fundamental difference between hierarchical mixtures of experts (HMEs) and oblique decision trees. HMEs use softmax units instead of decision units in non-terminal nodes, so HMEs only approach oblique decision trees in the theoretical limit. Hence, HMEs realize continuous functions instead of piece-wise constant functions as oblique decision trees.
>
> There are some practical advantages of using oblique decision trees over HMEs-style (soft) decision trees. For example, it requires a full tree traversal for HMEs to make a prediction in O(2^M) where M denotes the tree depth, while it only takes O(M) for oblique decision trees to make a prediction. Moreover, being piece-wise constant allows us to tractably compute some useful inference problems like “what is the feasible region of input space that leads to a specific prediction value / diagnosis”.
>
> Nevertheless, we can still do an empirical comparison with HMEs. We use the HME implementation with the most stars on GitHub: https://github.com/AmazaspShumik/Mixture-Models
>
> Even with parallel computing on 16 CPUs, the implementation is still prohibitively slow due to the inherent complexity of the HME learning algorithm (exponential to tree depth), and we can only obtain some early results on small datasets. The experiment protocol is the same. We tune the depth in {2,3,...,12} by validation set, and report the testing performance of the tuned model.
>
> We report (mean ± std) of testing AUC score over multiple runs.
> Bace:
> HME (4 runs):    0.706 ± 0.009
> LCN (10 runs):   0.839 ± 0.013
> ALCN (10 runs): 0.854 ± 0.007
>
> Sider:
> HME (1 run):      0.582 ± 0.000
> LCN (10 runs):   0.624 ± 0.044
> ALCN (10 runs): 0.653 ± 0.044
>
> Empirically, HME performs much worse than the proposed LCN and ALCN models.
>
> It takes 4.5 hours to train an HME on the Bace dataset with depth = 12, and it takes 42.3 hours on the HIV dataset with depth = 6, so we cannot report the complete experiments for the other datasets during rebuttal. To see the exponential time complexity:
>
> HMEs training time on the HIV dataset.
> Depth = 3:   5.0 hours
> Depth = 4:  11.1 hours
> Depth = 5:  23.7 hours
> Depth = 6:  42.3 hours

---

> > ### Comment · AnonReviewer2 · 2019-11-13
> > **Good Results**
> >
> > I appreciate the authors for clarifying my concern.
> > It seems LCN/ALCN can be promising alternatives of HMEs.
> >
> > [Just a small note]
> > Is it possible to train classical decision trees (i.e. only axis-aligned splittings) using LCN?

---

> > > ### Author Response · Authors · 2019-11-14
> > > **Re: Good Results**
> > >
> > > Thank you for the feedback.
> > >
> > > We agree that the setting is interesting. However, translated into our framework, the ReLU network (f_\theta) that yields axis-aligned splits would be somewhat simple or contrived. It is not clear for now whether the LCN approach offers any real advantage in this scenario. LCNs seem much better suited for the harder problem of oblique tree induction.

---

### Official Review · AnonReviewer1 · 2019-10-29
**Official Blind Review #1**

**Rating:** 6

**Review:**

This paper proposes locally constant network (LCN), which is implemented via the gradient of piece-wise linear networks such as ReLU networks. The authors built the equivalence between LCN and decision trees, and also demonstrated that LCN with M neurons has the same representation capability as decision trees with 2^M leaf nodes. The experiments conducted in the paper disclose that training LCN outperforms other methods using decision trees.

The detailed comments are as follows:

1) The idea of LCN is very interesting, and the equivalence to decision trees is also very valuable, as it provides interpretability and shines light on new training algorithms.

2) The derivation of LCN and the equivalence is clear. The analysis based on the shared parameterization in Section 3.5 is helpful to understand why LCN with M neurons could be of equal capability to decision trees with 2^M leaf nodes.

3) One weakness is that the performance of ELCN seems to be very close to RF, as shown in Table 2.

I am not sure whether some similar ideas to LCN have been explored in the literature. But the topic studied in this work is very valuable, which connects deep neural networks and decision trees.


**Experience Assessment:**

I do not know much about this area.

**Review Assessment: Checking Correctness Of Derivations And Theory:**

I carefully checked the derivations and theory.

**Review Assessment: Checking Correctness Of Experiments:**

I assessed the sensibility of the experiments.

**Review Assessment: Thoroughness In Paper Reading:**

I read the paper at least twice and used my best judgement in assessing the paper.

---

> ### Author Response · Authors · 2019-11-10
> **Official response**
>
> We thank the reviewer for the insightful comments and suggestions. Please see our response to a common comment above.

---

### Official Review · AnonReviewer3 · 2019-10-31
**Official Blind Review #3**

**Rating:** 3

**Review:**

*Summary*
This paper leverages the piecewise linearity of predictions in ReLU neural networks to encode and learn piecewise constant predictors akin to oblique decision trees (trees with splits made on linear combinations of features instead of axis-aligned splits). The core observation is that the Jacobian of a ReLU network is piecewise constant w.r.t to the input. This Jacobian is chosen to encode the hard splits of a decision tree. The paper establishes an exact equivalence between decision trees and a slightly modified form of the locally constant networks (LCN). The LCN used for experiments is slightly relaxed to allow for training, including "annealing" from a the softplus nonlinearity to ReLU during training, adding one or more output layers to perform the final prediction, and training with connection dropout. Experiments show LCN models outperform existing methods for oblique decision trees, but ensembles are often matched or outperformed by random forests.

*Rating*
Perhaps the greatest attribute of decision trees is utter simplicity. (The second best attribute the out-of-the-box competitive accuracy of tree ensembles on a wide variety of problems.) An argument to be made for this paper is that it leverages the machinery of learning DNNs to learn more powerful, oblique tree-like models. The counterpoint is that despite the added complication, it's still often beaten by ensembles of CART trees. Overall, the idea is clever, the presentation could be improved slightly, and the experiments raise existential questions for this kind of work. My current rating is weak reject.

(1) It's difficult to know how LCNs should be compared to traditional decision trees, with accuracy, number of parameters, prediction speed, and training time/parallelism as viable components. The paper focuses almost exclusively on accuracy, while cross-validating over model sizes and other hyperparameters. This is a reasonable choice, though a discussion of model size and prediction speed would be welcome. I do have two significant questions about the experiments:

(2) It seems unfair that LCN has access to one or more hidden layers between the splits and the final output, denoted g_\phi. Would competing decision tree models improve with such a layer learned and appended to the final tree? Would LCN suffer from using a tabular representation like the others?

(3) Despite the assertion that these are datasets that necessitate tree-like predictors, the LLN method outperforms LCN and the trees on 4/5 datasets and is competitive with ensemble methods. While not explicitly stated, am I correct that LLN is essentially a traditional ReLU-network? If high accuracy is the goal, then why should I go to the trouble of training LCN when a traditional DNN is better. And if a tree is needed, then LCNs should be evaluated on more than just accuracy.

(4) LCNs seem to present a less bulky alternative to e.g. Deep Neural Decision Trees (https://arxiv.org/abs/1806.06988), but that work should be cited and discussed

(5) The proof sketch in Section 3.6 of the equivalence between the "standard architecture" and decision trees is difficult to understand and not convincing. (On second reading I noticed the subtle vector "\mathbf 0" indicating that all entries of "\grad_x a^i_1" are zero. Some further exposition and enumeration of steps would clear up confusion.)

(6) Overall the presentation is reasonable, other than the notes below. I did find myself searching back over the (dense) notation section and following sections looking for definitions of variables and terms used later. Consider better formatting (e.g. more definitions in standalone equations), strategic pruning of some material to make it less dense, and repeating some definitions in line (e.g. see below for "p7:... remind the reader").

*Notes*
(Spelling typos throughout; most are noted below)
p3: clarify in 3.1/3.3 that L is the number of outputs
p4: "interpred"
p5: "aother"
p5: Theorem 2 proof: note that the T/T' notation is capturing left/right splits
p5: "netwoworks"
p5: "Remark 5 is important for learning shallow...": should "shallow" be "narrow" instead?
p7: in the first paragraph, remind the reader of the definitions of õ^M and J_x ã^M
p7: "Here we provide a sketch of [the] proof"
p7: "unconstraint" should be "unconstrained"
p7: "...can construct a [sufficiently expressive] network g_\theta"
p7: "simlify"
p9: Table 2: instead of "2nd row", ..., use "1st section", ...; also consider noting which methods are introduced in this paper
p9: Figure 2: text is too small


**Experience Assessment:**

I have read many papers in this area.

**Review Assessment: Checking Correctness Of Derivations And Theory:**

I carefully checked the derivations and theory.

**Review Assessment: Checking Correctness Of Experiments:**

I carefully checked the experiments.

**Review Assessment: Thoroughness In Paper Reading:**

I read the paper thoroughly.

---

> ### Author Response · Authors · 2019-11-10
> **Official response - 1**
>
> We thank the reviewer for the insightful comments, suggestions and questions. Please also see our response to a common comment above.
>
> (1) How to compare LCNs with oblique decision trees: we may consider three different factors.
>
> 1. Theoretical capacity: it is analyzed in Sec. 3.4-3.5.
> 2. Model training:
>     (a) Optimization performance: it is analyzed in Figure 2. LCNs is competitive with the state-of-the-art method.
>     (b) Training speed:
>         (i) If assuming matrix multiplication and a batch can be fully parallelized, the time complexity is O(MT (N/B) ), where M is the tree depth, T is the number of epochs, N is the number of data, and B is the batch size. One can always tune the batch size and the number of epochs (and even optimizers) to optimize training speed. We only tuned the validation performance in our experiments.
>         (ii) The time complexity for traditional methods is hard to analyze precisely, since the number of data in each split highly depends on the dataset.
>         (iii) In our experiments, the training time with depth = 12 on the HIV dataset is 8.8 min for HHCART / 8.6 min for TAO / 18.0 min for LCN with batch size = 64.
>     (c) Model size:
>         (i) The space complexity of decision nodes in oblique decision trees is O(2^M D). The complexity of the ReLU network f in LCNs (which yields decision nodes) is O(M D).
>         (ii) The space complexity of leaf nodes v / the table g in oblique decision trees / canonical LCNs is O(2^M). The complexity of g_\phi in the standard LCN depends on how it is parameterized.
>         (iii) In our experiments, LCN with depth = 12 and batch size = 64 costs 541Mb GPU memory on the HIV dataset.
>
> 3. Model testing:
>     (a) Generalization performance: it is analyzed in Table 2 and Figure 2. LCNs outperform traditional approaches by a large margin.
>     (b) Testing speed / model size:
>         (i) Since we can explicitly convert an LCN to an oblique decision tree, and vice versa, the prediction speed / model size does not really matter.
>         (ii) In our experiments, the prediction time with depth = 12 on the HIV testing set is 30.4 sec for HHCART / 2.6 sec for TAO / 0.4 sec for LCN. We again use batch size = 64 on GPU for LCN. The difference between HHCART and TAO is largely due to how the oblique cuts are implemented (numpy linear projection vs. LibLinear).

---

> ### Author Response · Authors · 2019-11-10
> **Official response - 2**
>
> (2) In short, we have the following ordering of expressiveness *given a fixed depth*: oblique decision trees > canonical (tabular) LCNs >= standard LCNs.
>
> Note that each activation pattern of f yield a *constant* Jacobian, so we can write the Jacobian as Jacobian(o(x)), where o is the activation pattern of x.
>
> Then, standard LCNs can be written as the mapping:
>         g_\phi(Jacobian(o(x))).
> In contrast, canonical (tabular) LCNs yield the mapping:
> 	g(o(x)).
> Hence, the tabular case cannot be less powerful than using embeddings (Jacobians), since we can always set the table as g(  ) = g_\phi(Jacobian(  )).
>
> By the proof of Theorem 3, we can again transform such tabular LCN to a decision tree with the same depth, so the canonical LCNs are not more powerful than oblique decision trees. Furthermore, due to how the locally linear regions of f can be partitioned, canonical LCNs are strictly less powerful than oblique decision trees given a fixed depth M as proved in Sec. 3.5.
>
> Hence, we obtain the claimed ordering. In fact, by Theorem 8 (updated, originally given as a sketch of proof), we can even prove standard LCNs = canonical LCNs.
>
>
> (3) Yes, LLNs are ReLU networks. The point of including LLN is to compare it with ALCN, since both are continuous functions and thus much more powerful than tree methods. We show that ALCN often outperforms LLN.
>
> If high accuracy is the only goal, currently random forests work well for these chemical/tabular data, outperforming neural networks with ReLU activations by a large margin. However, please see our general response above for why LCN still provides new insights in this setting.
>
> If a tree model is needed, clearly LCNs are much better than traditional oblique decision trees: we can often get 10% absolute improvements in testing AUC. Since LCN can always be explicitly converted to an oblique decision tree, the only difference *in testing time* between oblique decision trees and LCNs is the accuracy. Other factors are discussed in (1).
>
> (4) Thank you for the reference. It seems a relevant paper that also uses deep networks to learn decision trees. The paper focused on axis-parallel decisions (traditional decision trees), while we work on oblique decisions. We will cite and discuss the paper in the camera-ready version.
>
> (5) Thank you for the feedback. The sketch of the proof is now replaced by a formal theorem with proof in Appendix A.2.
>
> (6) and *Notes*: Thank you for the comments. The typos have been addressed, and we will improve the other presentation issues in later revision.
>
> Re "Remark 5 is important for learning shallow..." Thank you for pointing this out. Here we meant that given a fixed number of neurons, the one neuron per layer setting is the most powerful architecture.
>
> Minor technical clarification: we “anneal” LCNs during training, but the LCNs used in testing time are indeed piece-wise constant (not relaxed). Adding any number of layers mapping from the Jacobian to the prediction does not affect the piece-wise constant nature.

---

### Author Response · Authors · 2019-11-10
**Official response to a common comment**

(R1 and R3) The performance of ELCN seems to be very close to RF:

Yes, the performance of ELCN is indeed very close to RF in the property prediction datasets. Please note that RFs were already tailored and chosen by others for these datasets since they can work particularly well in these cases. Our ELCNs represent the first cut demonstration of LCN ensembles, and they already outperform traditional counterparts, GBDT.

We could only compare to oblique decision trees and fully ignore the ensemble setting, but we believe that including the comparison is helpful for highlighting its potential given the improvement from GBDT to ELCN.

Finally, we would like to emphasize that the main contribution of this paper is the LCN model and how/why we may use it to obtain a better oblique decision tree than traditional approaches. The continuous extension (ALCN) and ensemble (ELCN) are nice extensions, but they should not be evaluated as the focus of this paper.

---

### Decision · Program_Chairs · 2019-12-19

**Decision:**

Accept (Poster)

**Comment:**

This paper leverages the piecewise linearity of predictions in ReLU neural networks to encode and learn piecewise constant predictors akin to oblique decision trees. The reviewers think the paper is interesting, and the idea is clever. The paper can be further improved in experiments. This includes comparison to ensembles of traditional trees or (in some cases) simple ReLU networks. Also  the tradeoffs other than accuracy between the method and baselines are also interesting.